# Constitutive Expression of a Cytotoxic Anticancer Protein in Tumor-Colonizing Bacteria

**DOI:** 10.3390/cancers15051486

**Published:** 2023-02-27

**Authors:** Phuong-Thu Mai, Daejin Lim, EunA So, Ha Young Kim, Taner Duysak, Thanh-Quang Tran, Miryoung Song, Jae-Ho Jeong, Hyon E. Choy

**Affiliations:** 1Department of Microbiology, Chonnam National University Medical School, Gwangju 61468, Republic of Korea; 2Odysseus Bio, Basic Medical Research Building, Chonnam National University Medical College, 322 Seoyangro, Hwasun, Jeonnam 58128, Republic of Korea; 3Department of Biotechnology, Vietnam—Korea Institute of Science and Technology, Hanoi 100000, Vietnam; 4Division of Biomedical Convergence, College of Biomedical Science, Kangwon National University, Chuncheon 24341, Republic of Korea; 5Department of Bioscience and Biotechnology, Hankuk University of Foreign Studies, Yongin 17035, Republic of Korea

**Keywords:** *Escherichia coli*, *Salmonella enterica* serovar Gallinarum, anticancer protein expression, host response, bacterial cancer therapy

## Abstract

**Simple Summary:**

This study examined the biodistribution of *Escherichia coli* and an attenuated strain of *Salmonella enterica* serovar Gallinarum with defective ppGpp synthesis after injection into tumor-bearing mice through the tail vein. Bacteria targeting tumor tissues, but not those in the liver and spleen, were metabolically active and proliferated substantially. Recombinant bacteria derived from the attenuated *Salmonella enterica* serovar Gallinarum that constitutively expressed transforming growth factor α (TGFα) fused to a modified *Pseudomonas* exotoxin A (PE38) showed marked antitumor effects on tumor-bearing mice without any notable systemic toxicity.

**Abstract:**

Bacterial cancer therapy is a promising next-generation modality to treat cancer that often uses tumor-colonizing bacteria to deliver cytotoxic anticancer proteins. However, the expression of cytotoxic anticancer proteins in bacteria that accumulate in the nontumoral reticuloendothelial system (RES), mainly the liver and spleen, is considered detrimental. This study examined the fate of the *Escherichia coli* strain MG1655 and an attenuated strain of *Salmonella enterica* serovar Gallinarum (*S.* Gallinarum) with defective ppGpp synthesis after intravenous injection into tumor-bearing mice (~10^8^ colony forming units/animal). Approximately 10% of the injected bacteria were detected initially in the RES, whereas approximately 0.01% were in tumor tissues. The bacteria in the tumor tissue proliferated vigorously to up to 10^9^ colony forming units/g tissue, whereas those in the RES died off. RNA analysis revealed that tumor-associated *E. coli* activated rrnB operon genes encoding the rRNA building block of ribosome needed most during the exponential stage of growth, whereas those in the RES expressed substantially decreased levels of this gene and were cleared soon presumably by innate immune systems. Based on this finding, we engineered ΔppGpp *S.* Gallinarum to express constitutively a recombinant immunotoxin comprising TGFα and the *Pseudomonas* exotoxin A (PE38) using a constitutive exponential phase promoter, the ribosomal RNA promoter rrnB P1. The construct exerted anticancer effects on mice grafted with mouse colon (CT26) or breast (4T1) tumor cells without any notable adverse effects, suggesting that constitutive expression of cytotoxic anticancer protein from rrnB P1 occurred only in tumor tissue.

## 1. Introduction

Bacterial cancer therapy relies on the inherent traits of certain facultative anaerobic bacteria that are capable of intratumoral penetration and localization in hypoxic areas presumably because of chemotaxis toward molecules produced by tumors and the immune-privileged environment of tumors [1,2,3]. Among gram-negative anaerobes, *Salmonella* spp. has prevailed as a therapeutic candidate because it is amenable to genetic manipulation and can trigger an immune response [4,5,6,7,8,9]. The initial response to the bacterial colonization of tumors is the secretion of the proinflammatory cytokine TNFα by innate immune cells, which causes a hemorrhage in the tumor and the formation of large necrotic regions [10,11]. The hemorrhage and necrosis formation induce tumor growth retardation. This is followed by the strong adjuvant effect of tumor-specific T cells that are activated by the colonizing bacteria [12,13,14,15,16]. CD8^+^ cytotoxic T cells are the main type of cells that counteract tumor growth.

Tumor-colonizing bacteria can be used as a delivery system for therapeutic molecules that promote tumor elimination. However, this type of application is problematic for pathogenic *Salmonella* spp. that can invade animal cells and reside within membrane-bound compartments, namely, *Salmonella*-containing vacuoles (SCVs) [17,18]. The transportation of anticancer proteins expressed by SCV-bound *Salmonella* into the cancer cell cytosol is another potential complication; therefore, *Salmonella* should be prevented from invading host cells. Another challenging aspect of this approach is the specific targeting of cytotoxic anticancer proteins to solid tumors but not to the reticuloendothelial system (RES), the liver and spleen, where most of the bacteria are initially trapped [19,20,21]. To overcome this problem, bacteria are often genetically engineered to express a specific cytotoxic anticancer gene only when they accumulate in tumor tissues after being eliminated from the RES to prevent damage to these organs by toxic therapeutic molecules. We previously used the P_BAD_ promoter from the *E. coli* arabinose operon, which can be activated by L-arabinose to induce the selective expression of cytotoxic anticancer proteins, although multiple injections of the inducer can cause problems for patients [20,22,23,24,25].

In this study, we determined the fate of the *E. coli* K12 strain MG1655 and an attenuated strain of *Salmonella enterica* serovar Gallinarum (*S.* Gallinarum), in which ppGpp synthesis was disabled after intravenous injection into tumor-bearing mice. The results showed that the bacteria that colonized the tumor tissue proliferated vigorously, whereas those in the RES were metabolically inert and were cleared in a short time by the innate immune cells. The ΔppGpp strain of avian-specific *S.* Gallinarum, the antitumor characteristics of which will be described in a separate manuscript, is defective in host cell invasion or intracellular survival [26]. We engineered ΔppGpp *S.* Gallinarum to express an immunotoxin (TGFα-PE38, TP) using the constitutive exponential phase promoter, the ribosomal RNA promoter rrnB P1, and found that it has remarkable antitumor effects on tumor-bearing mice without causing any adverse effects.

## 2. Materials and Methods

### 2.1. Bacterial Strains, Plasmids, and Culture Conditions

Bacterial strains and plasmids are listed in Table 1. *E. coli* K-12 MG1655 and *S. enterica* serovar Gallinarum clinical isolate (SG4021) were used as wild-type strains and cultured in Luria Bertani broth (LB, Difco Laboratories, Franklin Lakes, NJ, USA). The ΔppGpp *S.* Gallinarum (SG4023) was constructed from SG4021 using the λ red system to disrupt the relA and spoT genes as described previously [26]. The ΔppGpp Δ*glmS S.* Gallinarum (SG4030) was constructed by p22 phage transduction as described previously [27] using 10% N-acetyl-D-glucosamine in LB medium. To monitor rrnB P1 promoter activity, the prrnBP1-gfpOVA plasmid was constructed by Gibson assembly as follows: first, the rrnB P1 promoter containing Fis-binding sites in the upstream activation region (−154–+3) was amplified from the *E. coli* K-12 MG1655 chromosome [28] and cloned into a reporter gene (*gfpOVA*) by replacing the katG promoter sequence in the pkatG::gfpOVA plasmid [29]. The rrnB P1 promoter replaced the *araBAD* promoter sequence in the pSEC-TGFα-PE38 plasmid [25] with the specific primer set listed in Table 2, generating the prrnBP1-psp-TP plasmid. The balanced-lethal system based on the glmS gene was introduced into prrnBP1-psp-TP in the claI site to maintain the plasmid in vivo [23,27]. Plasmids were confirmed via DNA sequencing (Macrogen, Seoul, Republic of Korea). Each plasmid was introduced into *E. coli* by heat shock or *Salmonella* by electroporation. Bacteria strains carrying the plasmid were grown in LB medium with 1% NaCl at 37 °C with vigorous shaking. When necessary, ampicillin (Sigma-Aldrich, Darmstadt, Germany) was added at a concentration of 100 µg/mL. To identify amino acids required for growth, wild-type *S.* Gallinarum and ΔppGpp *S.* Gallinarum were cultured on M9 Minimal Medium plates (Welgene Precision Solution^TM^, Gyeongsan, Republic of Korea) supplemented with glucose (0.2 g/mL), thiamine (5 mg/mL), magnesium sulfate (1 M), calcium chloride (1 M), and a mixture of 19 amino acids (100 µg/mL, each) with 1 omitted from 20 essential amino acids.

### 2.2. Cell Lines and Animal Experiments

Female BALB/c mice (6–8 weeks, 18–20 g) were obtained from Orient Bio (South Korea). CT26 colon cancer cells and 4T1 murine mammary carcinoma cells were purchased from ATCC Korea and cultured in high-glucose DMEM supplemented with 10% fetal bovine serum and 1% penicillin-streptomycin. Cells (1 × 10^6^) in 30 µL of 1× PBS were subcutaneously injected into the right thigh of each mouse. Bacterial injections were executed when tumors reached a size of 100–150 mm^3^. For confirmation of rrnB P1 promoter activity in the tumor targeted bacteria, *E. coli* K-12 MG1655 and ΔppGpp *S.* Gallinarum, mice carrying CT26 xenografts were intravenously injected with bacteria carrying prrnBP1-gfpOVA (1 × 10^8^ colony forming units [CFU]/mouse). To examine the antitumor effects of the TP immunotoxin, the mice grafted with CT26 and 4T1 cells were injected with ΔppGpp Δ*glmS S.* Gallinarum carrying prrnBP1-psp-TP through the tail vein. Tumor size was determined by measuring the length, width, and height of each tumor every 2 days after the injection (V = length × width × height × 0.5). For bacterial distribution in vivo, solid tumors and other organs were extracted from mice and homogenized in 1× PBS using a homogenizer (IKA, Ultra–Turrax T10). The bacteria counting method was previously described [25]. All mouse experiments were performed by following the guidelines of Chonnam National University–Institutional Animal Use and Care Committee (CNU IACUC-H-2020-7). The protocol requires sacrifice of the mice when the implanted tumor volume reaches > 1500 mm^3^.

### 2.3. Bacterial RNA and cDNA Library Preparation

For in vivo experiments, excised tissues were stored at −80 °C in 1 mL tubes containing RNA protection reagent (Qiagen, Hilden, Germany). For RNA isolation, 50–100 mg tissue was homogenized in 1 mL Trizol (Gene All, Seoul, Republic of Korea, RiboEx, cat. no. 301–001). RNA extraction procedures were performed according to the manufacturer’s recommendations. RNA samples were treated with *DNase I* to minimize genomic DNA contamination, and RNA integrity and quantity were confirmed by agarose gel electrophoresis and NanoDrop (Eppendorf, Tokyo, Japan, BioSpectrometer). cDNA was synthesized from 1–5 µg total RNA using reverse transcriptase with random hexamer primers (Enzynomics, Daejeon, Republic of Korea, TOPscript^TM^ cDNA Synthesis Kit, cat. no. EZ005S).

### 2.4. Quantitative Polymerase Chain Reaction (Real-Time PCR)

The qPCR mixtures (20 µL) consisted of the template cDNA (30 ng), a primer set (0.25 µM, each), and qPCR 2× PreMix (10 µL) (Enzynomics, TOPreal^TM^ qPCR 2× PreMix, SYBR Green with lox ROX). To measure the expression level of gfp derived by the rrnB P1 promoter, cDNA was amplified with the forward primer 5′-GCAGACCATTATCAACA AAATACTCC-3′ and the reverse primer 5′-CTTTCGAAAGGGCAGATTGTGT-3′. As a reference gene, rpoB was used for qPCR using the forward primer 5′-CGCGTATGTCCAATCGAAA-3′ and the reverse primer 5′-GAGTCTCAAGGAAGCC GTATTC-3′ for *E. coli*, and the forward primer 5′-GCGTCTCAAGGAAGCCATATTC-3′ and the reverse primer 5′-GTCGCGTATGTCCTATCGAAAC-3′ for *S.* Gallinarum. The analysis was performed with a Rotor-GenQ real-time PCR system (Qiagen, Rotor-GenQ series software, v.2.2.3). The 40 PCR cycles were conducted as follows: initial denaturation at 95 °C for 15 min, denaturation at 95 °C for 10 s, annealing at 60 °C for 15 s, and elongation at 72 °C for 15 s. The cycle threshold (Ct) values obtained from amplifying the cDNA of the gfp gene were normalized to Ct values of the reference gene rpoB by the 2^−ΔΔCt^ method in triplicate.

### 2.5. RNA Sequencing Analysis

At 1 and 3 days after *E. coli* injection, total RNA was extracted from the indicated organs and tumor tissues as described above. RNA quantification and purity assessment were performed using a 2100 Bioanalyzer (Agilent Technologies, Waldbronn, Germany). A sequencing library was prepared with 1 μg total RNA for each sample using the Illumina TruSeq Stranded Total RNA LT Sample Prep Kit (Illumina, San Diego, CA, USA). The resulting cDNA libraries were sequenced using the NovaSeq platform (Illumina), generating approximately 2.78 billion paired end reads of 101 nucleotides in length. To obtain high-quality clean reads from the sequenced raw reads, quality-based filtering and trimming were performed using Trimmomatic (v.0.36) with the following parameters: ILLUMINACLIP:TruSeq3-PE-2.fa:2:30:10LEADING:3TRAILING:3SLIDINGWINDOW:4:15 MINLEN:36. To analyze the *E. coli* transcriptome in the mouse liver and tumor tissues, clean reads were mapped to the mouse reference genome (mm10) using HISAT (v.2.1.1) with default parameters. Then, unmapped reads were extracted using Samtools (v.1.9) and remapped to the *E. coli* K-12 MG16555 reference genome. To identify the read coverage of the rrnB operon, the alignment results were input to the Samtools depth command with the following range of *E. coli* chromosome 4: 165,658–4,172,756. The number of total RNA-seq reads was used for normalization.

### 2.6. Bacterial Division Analysis

Flamma^®^ Fluors 552 N-hydroxysuccinimide (NHS) ester, a labeling fluorescent dye, was purchased from BioActs (Incheon, Republic of Korea, cat. no. PWS1122) and dissolved in DMSO (Biosesang, DR1022-500-00). The overnight culture of ΔppGpp *S.* Gallinarum (1 × 10^9^ CFU/mL) in 2 mL 1× PBS was conjugated with the above fluorescent dye (final concentration: 100 µg/µL) under slow-speed rotation at room temperature overnight. The stained bacteria were subcultured in LB broth (ratio: 1:100) in a 37 °C shaking incubator. Bacterial growth at A_600_ and red fluorescence intensity at λexcitation = 550 nm and λemission = 610 nm were measured every hour using a spectrophotometer (Shimazu, Kyoto, Japan, UV-1800) and a fluorometer (Thermo Scientific, Waltham, MA, USA, VarioskanLux).

For in vivo analysis, the NHS-conjugated bacteria were injected into CT26-grafted mice through the tail vein (*n* = 5 per group, 1 × 10^8^ CFU/mouse). To examine bacterial conditions in mice, the tumors, livers, and spleens were excised at the indicated times after bacterial infection and processed for the detection of bacteria and F4/80^+^ macrophages by confocal microscope.

### 2.7. Immunofluorescent Staining and Confocal Microscope

NHS-conjugated *Salmonella* was collected at the indicated times, washed with 1× PBS, fixed with 3.9% formaldehyde, and placed on glass slides. The samples were incubated with an anti-*Salmonella* antibody (antirabbit, Abcam (Cambridge, UK), ab35165, 1:50) overnight at 4 °C. After washing with 1× PBS, the samples were treated with the secondary antibody, Alexa Fluor^®^ 488-conjugated goat antirabbit antibody (Invitrogen (Waltham, MA, USA), REF. A11008, 1:100).

In vivo analysis of tumor targeting bacteria was performed using the isolated organs from tumor-bearing mice treated with each bacterial strain. The organs from the mice were fixed with 3.9% formaldehyde overnight at room temperature and embedded in 20% sucrose (Sigma-Aldrich) to remove formaldehyde. Tissues were then frozen in OCT compound (Optimal Cutting Temperature, Tissue-Tek, Torrance, CA, USA) and sliced into 7 µm-thick sections using a microtome (Thermo Scientific, Cryostat Microm HM525). To remove the OCT compound, the slices were dried for 15 min at room temperature, washed three times with 1× PBS, and fixed with absolute acetone. The slides were incubated in 1× PBS containing rabbit anti-*Salmonella* (Abcam, ab35165, 1:100) and rat anti-F4/80^+^ macrophage (Abcam, ab6640, 1:100) primary antibodies overnight at 4 °C, followed by washing and incubation in 1× PBS with Alexa Fluor^®^ 633-conjugated goat antirabbit antibody (Life Technologies, Carlsbad, CA, USA, REF. A21071, 1:100) and Alexa Fluor^®^ 488-conjugated goat antirat antibody (Life Technologies, REF. A11006, 1:100) for 2 h at room temperature. Nuclei were stained with DAPI for 10 min at room temperature (Invitrogen, 1:1000), and slides were covered with antifade DAPI (Invitrogen, REF. P36935). Samples were visualized using a confocal microscope, and images were acquired using ZEN blue edition 2.6 V7.0.

### 2.8. Western Blot Analysis

To examine the expression of the TP protein in vitro, overnight cultures of ΔppGpp *S.* Gallinarum (SG4023) and ΔppGpp Δ*glmS S.* Gallinarum harboring the plasmid prrnBP1-psp-TP (SMP4003) were subcultured in LB broth (1:100) and grown for 7 h. At the indicated time points, bacterial pellets were collected and sonicated in 1× PBS. The supernatants were collected and filtered through 0.2 µm filters (GVS Filter Technology, USA). For animal experiments, SMP4003 (1 × 10^8^ CFU/mouse) was injected into the mice grafted with CT26 tumors through the tail vein when tumors reached 130–150 mm^3^. Tumors were excised at the indicated days and homogenized in 1 mL RIPA buffer (Intron Biotechnology, Seongnam, Republic of Korea) containing 1× Protease & Phosphatase Inhibitor Cocktail and 1× EDTA (Thermo Scientific). The filtered supernatants were mixed with 6× SDS and boiled at 95 °C for 10 min. The proteins were loaded onto 10% SDS-PAGE gels and transferred to nitrocellulose membranes (GE Healthcare, Solingen, Germany, cat no. 10600002). TP expression was determined by western blot analysis using a primary polyclonal antibody against PE38 (Sigma-Aldrich, P2318, antirabbit, 1:5000). Spontaneous bacterial lysis was examined by detecting GroEL using a specific antibody (Sigma-Aldrich, G6532, antirabbit, 1:5000). The level of β-actin was determined using a specific rabbit polyclonal antibody (Abcam, ab8227, 1:2000). Membranes were incubated with primary antibodies diluted in 5% skim milk in TBST at 4 °C overnight, followed by incubation in mouse antirabbit IgG-HRP (Santa Cruz Biotechnology, Dallas, TX, USA, sc-2357, 1:2000) for 1 h at room temperature. The proteins were visualized using ECL (Thermo Scientific, REF. 32209).

### 2.9. Statistical Analysis

Data were analyzed using GraphPad Prism v.8.0.2 software. The differences between the mean values of the two groups were analyzed using the unpaired two-tailed Student’s *t*-test. Two-way analysis of variance (ANOVA) was used for time-course studies. The survival rates are shown in Kaplan–Meier curves with log-rank (Mantel-Cox) test. Differences with *p* < 0.05 indicated statistical significance.

## 3. Results

### 3.1. Fate of E. coli Injected into Tumor-Bearing Mice

The fate of bacteria injected into tumor-bearing mice through the tail vein was examined using a common laboratory strain of *E. coli*, MG1655. *E. coli* MG1655 (1 × 10^8^ CFU/mouse) was injected into BALB/c mice bearing CT26 colon cancer xenograft tumors. At the indicated times after the injection, bacterial numbers were counted in the RES, in the liver and spleen, and in tumors using plating methods (Figure 1A). At 2 h after the injection, there were approximately 1 × 10^7^ CFU in the RES, and this number decreased gradually in a time-dependent manner, reaching approximately 5 × 10^4^ CFU at 120 h. The bacterial number in tumors was 1 × 10^4^ CFU at 2 h after the injection, and this increased to approximately 5 × 10^8^ CFU at 72 h. This result indicates that although ~0.01% of the injected bacteria accumulated in tumor tissues initially, the immunocompromised environment allowed substantial proliferation of those bacteria, whereas those in the RES were cleared presumably by phagocytic immune cells (see below). Among the most activated genes the most highly induced was the rrnB operon, consisting of the transcription unit rrsB-gltT-rrlB-rrfB encoding the three major rRNA building blocks of ribosomes [32]. The expression profile of the rrnB operon (number of reads) was determined by RNA sequencing (Figure 1B). The normalized read coverages in the liver and tumor were 58,220 and 1,148,213 reads at 1 dpi, respectively. At 3 dpi, these values changed drastically because of a shortfall of reads in the liver, showing 67 and 1,780,236 reads for the liver and tumor, respectively. We hypothesized that the reads on day 1 in the liver were remnants of those from overnight culture. By day 3, the bacteria in the liver were perishing as rrnB expression ceased, whereas those in the tumor proliferated. To confirm these findings, we measured the activity of the rrnB P1 promoter, which is the major promoter driving the rrnB operon. This promoter is active during the early exponential phase of growth, when ribosomes are needed most, and declines sharply thereafter during the stationary phase [33]. Three Fis-binding sites in the upstream activation region are responsible for the activation of the rrnB P1 promoter (Figure 1C) [28]. A gene reporter system was constructed using the unstable GFP variant gfpOVA [29], which was cloned downstream of rrnB P1 in pBR322, generating prrnBP1-gfpOVA. *E. coli* transformed with this plasmid were used to monitor rrnB P1 activity. During growth in vitro (Appendix A), fluorescence intensity determined at 488–522 nm indicated activation of rrnB P1 during the exponential phase of growth in LB medium, in agreement with the results of qPCR analysis of gfpOVA structural RNA. Then, we attempted to determine rrnB P1 activity in the *E. coli* injected into tumor-bearing mice using a fluorescence microscope; however, this failed due to weak emission of fluorescence. Alternatively, we measured rrnB P1 activity by qPCR analysis of the gfpOVA structural RNA relative to rpoB RNA, which is maintained at constant levels (Figure 1D) [34]. The activity of rrnB P1 increased by up to 40-fold in the bacteria in tumor tissues at 120 h, whereas those in the liver and spleen decreased over time. This result supports that those bacteria in tumor tissues proliferated, whereas those in the RES perished. We did not quantify the rRNA from the genomic rrnB operon because there are seven rrn operons in *E. coli* with similar sequences, and the ribosomal RNAs that provide the foundation for ribosomes are extremely stable.

### 3.2. Fate of ppGpp-Defective S. Gallinarum Injected into Tumor-Bearing Mice

For bacteria-mediated cancer therapy, *Salmonella* spp. that trigger effective IL-1β/TNF-α-mediated immune responses in the tumor leading to tumor regression are preferred over *E. coli* [9]. An attenuated strain of avian host-specific *S. enterica* serovar Gallinarum was constructed by deleting relA and spoT, which encode enzymes that synthesize the bacterial signaling molecule ppGpp [26]. The ppGpp defect causes amino acid auxotrophy in *S. Typhimurium* and in *E. coli* [35]. We observed that the ppGpp-defective *S.* Gallinarum also required several amino acids to grow, including branched chain amino acids in addition to lysine and serine (Appendix A). The ΔppGpp strain of *S.* Gallinarum was attenuated by approximately 1000-fold in mice, which allowed injection of 10^8^ CFU/mouse, resulting in regression of various tumors grafted in mice (manuscript in preparation). In this study, we examined the fate of ΔppGpp *S.* Gallinarum after its injection into the tail vein of BALB/c mice bearing CT26 xenograft tumors. Similar to the *E. coli*, the bacterial counts in the tumor increased, whereas those in the RES decreased in a time-dependent manner (Figure 2). To obtain a clear picture of the fate of ΔppGpp *S.* Gallinarum injected into tumor-bearing mice, we measured cell division using bacteria that were cross-linked with the reactive form of a fluorescent dye (Flamma^®^ Fluors 552: NHS), which reacts readily with amine-modified oligonucleotides or amino groups of proteins on the bacterial surface [36,37]. The bacteria incubated with the dye initially emitted strong red fluorescent signals when excited with a 550 nm laser light, which were visible under a fluorescence microscope (Figure 2A, 0 h). These bacteria were diluted in fresh LB medium (1/50) and grown with vigorous aeration (Figure 2B). Fluorescent signals from the bacterial cultures were detected at the indicated times and cell number was also estimated by determining optical density (A_600_). As the bacteria divided, the fraction of red fluorescent bacteria decreased, disappearing after approximately 2 h (four generations assuming g = 30 min) (Figure 2A,B). BALB/c mice bearing CT26 xenografts were injected with the fluorescent bacteria, and samples of the RES and tumor tissues were collected at the indicated times for the measurement of bacterial numbers and fluorescent signals using a fluorescence microscope (Figure 2C,D). Red fluorescent bacteria were observed in the RES even at 72 h after the injection, whereas they were rarely detected in tumor tissues after 12 h, indicating that the bacteria in the tumor tissue divided and diluted out the fluorescent dye (Figure 2E). In this experiment, the same tissue samples were stained for F4/80^+^ macrophages, and the results showed that most of the bacteria in the RES were associated with macrophages, whereas those in the tumor were not (Appendix A). Lastly, the activity of the exponential phase promoter rrnB P1 in ΔppGpp *S.* Gallinarum was measured in vitro (Appendix A) and in vivo by qPCR analysis (Figure 2F). The activity of rrnB P1 in the tumor increased up to 72 h, whereas that in RES decreased over time. Taken together, these results suggest that the *S.* Gallinarum accumulating in tumor tissues proliferated, whereas those in the RES were cleared by phagocytic macrophages.

### 3.3. Expression of an Immunotoxin under the Control of the Ribosomal RNA Promoter (rrnB P1)

To determine whether the rrnB P1 promoter could drive the expression of cytotoxic anticancer proteins, the rrnB P1 promoter sequence was cloned in place of the *araBAD* promoter in pBAD24 [38], which was fused to the open reading frame of the immunotoxin TP [25,27]. This immunotoxin (TP) comprising TGFα and a modified *Pseudomonas* exotoxin A (PE38) derived from *Pseudomonas aeruginosa* was developed for the treatment of EGFR-expressing malignant tumors such as brain tumors [39,40,41]. PE38 acts by inactivating protein synthesis in mammalian cells [42,43]. PE38, which lacks an intrinsic cell-binding domain, binds to EGFR-expressing cancer cells via the TGFα moiety in the recombinant toxin. The TP protein is cytotoxic to EGFR-expressing tumor cells in vitro and in xenograft mouse models [25,44]. In this study, we used the ribosomal RNA promoter rrnB P1 to express TP constitutively. The psp secretion signal peptide composed of 32 amino acids [25] was fused in-frame to the N’ end of TGFα-PE38 in the plasmid named prrnBP1-psp-TP. In addition, the plasmid contained the glmS gene to ensure the maintenance of the plasmid by a balanced-lethal host vector system in GlmS^-^ mutant bacteria [27]. This mutant undergoes lysis when grown in the absence of N-acetyl-D-glucosamine (GlcNac) unless complemented by a plasmid carrying the glmS gene. The ΔppGpp strain of *S.* Gallinarum carrying the mutation in glmS was transformed with prrnBP1-psp-TP (SMP4003), grown in LB broth, and harvested at the indicated times to quantify the expression of TP. The bacterial cells and supernatant were separated and subjected to western blotting to detect TP expression (Figure 3A). Under the control of the rrnB P1 promoter, TP was expressed at high levels in the pellet in a constitutive manner, whereas it was detected in the supernatant at later time points, indicating that TP was secreted via the psp signal after a certain time. Next, we investigated the cytotoxic effect of the immunotoxin TP secreted from SMP4003 on cancer cell lines overexpressing EGFR, i.e., CT26 mouse colon carcinoma and 4T1 murine breast cancer cells (Appendix A) [25,45]. The bacteria were grown in LB medium and harvested when the culture entered the stationary phase. The cultures were centrifuged, and the supernatants were collected and concentrated. The CT26 and 4T1 cancer cell lines were treated with PBS or concentrated bacterial supernatant (1 µg protein). Approximately 70% of CT26 cells and 60% of 4T1 cells were killed after treatment with the supernatant of SMP4003 for 24 h. The supernatant from ΔppGpp *S.* Gallinarum only (SG4023) was included as a control and showed a moderate effect. These data indicate that TP released from SMP4003 is cytotoxic to these cancer cells.

The expression and secretion of TP from tumor targeted ΔppGpp *S.* Gallinarum carrying the prrnBP1-psp-TP (SMP4003) plasmid were evaluated in BALB/c mice grafted with mouse colon cancer CT26 cells (Figure 3B). At the indicated days after tail vein injection of the bacteria (1 × 10^8^ CFU), the grafted tumors were isolated and homogenized, and the supernatant was separated by centrifugation and filtered through 0.2 µm pores. The filtrate was analyzed for TP protein (43.3 kDa) expression by western blotting. TP was detected constantly throughout the course of the experiment from 1 to 5 dpi, suggesting that the protein was expressed constitutively from the rrnB P1 promoter and released from bacteria through the psp signal peptide. Next, we evaluated the antitumor effects of the immunotoxin on CT26 and mouse breast cancer 4T1 cells implanted into BALB/c mice (Figure 4). All mice received an intravenous injection of (i) PBS, (ii) ΔppGpp *S.* Gallinarum (SG4023) alone, or (iii) SG4030 carrying prrnBP1-psp-TP (SMP4003). Administration of SG4023 bacteria alone inhibited tumor growth for up to approximately 10 days compared with that in the PBS-treated group in both tumor models. Expression of the immunotoxin by the rrnB P1 promoter decreased tumor growth further. Average tumor size changes are shown in Figure 4A,D and the representative pictures are appeared in Figure 4B,E. The tumor sizes of individual mouse were also checked (Appendix A). Negative effects on the health of mice were rarely observed, and there was no significant difference in the body weight of mice between the groups (Appendix A). The mice treated with ΔppGpp *S.* Gallinarum expressing the immunotoxin survived 10–15 days longer than the mice treated with PBS or ΔppGpp *S.* Gallinarum alone (Figure 4C,F). Taken together, the results suggest that TP expressed from the constitutive rrnB P1 promoter in ΔppGpp *S.* Gallinarum effectively suppressed tumor growth without any additional manipulation and without causing side effects.

## 4. Discussion

In this study, we determined the fate of attenuated noninvasive ΔppGpp *S.* Gallinarum and the common laboratory strain of *E. coli* MG1655 after intravenous injection into tumor-bearing mice (10^8^ CFU). Approximately 10% of the injected bacteria were detected initially in the RES, whereas approximately 0.01% were in tumor tissues. The bacteria in the tumor tissue proliferated vigorously to up to 10^9^ CFU/g tissue, whereas those in the RES died off. (Figure 1 and Figure 2). The proliferation of bacteria in tumor tissues can be attributed to the unique immunosuppressive and biochemical environment of the tumor [46]. The rrnB P1 promoter in proliferating bacteria in tumor tissues was active, whereas that in the dying bacteria in the RES was not. The rrnB P1 promoter controls an operon that includes rrsB (16S rRNA), gltT (tRNA-glu), rrlB (23S rRNA), and rrfB (5S rRNA), which encode the three major rRNA building blocks of ribosomes [32]. In rapidly dividing bacteria, a large fraction of cellular energy and matter is devoted to the synthesis of ribosomes, accounting for 47% of the cell mass in *E. coli* when grown fast with a generation time of <30 min [47]. Because the rate of ribosome synthesis is determined solely by the availability of rRNA, the decrease observed suggests that bacteria in the RES ceased to grow and thus did not need ribosomes for de novo protein synthesis. It is likely that these bacteria were processed by phagocytic leukocytes, i.e., polymorphonuclear leukocytes (PMNs or neutrophils) and mononuclear phagocytes (monocytes, macrophages, and dendritic cells), providing a front line of defense against bacterial infection [48]. In this study, we showed that most of the bacteria in the RES were associated with phagocytic macrophages (Appendix A). These innate immune cells promote bacterial clearance through phagocytosis, generation of reactive oxygen and nitrogen species, extracellular trap formation, and production of proinflammatory cytokines [48]. The results presented in this study suggest that the confined expression of therapeutic payloads using a controllable system could be replaced by a constitutive expression system (Figure 3 and Figure 4). Metabolically inert bacteria that died out in macrophages detected in the RES discontinued protein synthesis, and leakage of therapeutic cargo into the circulation would thus be impossible. Several inducible promoter systems have been developed for the controlled expression of therapeutic cargo, including an *E. coli* promoter (pBAD) inducible with L-arabinose [20,49,50] and a tet promoter inducible with tetracycline [24,49]. Any transgene under these promoters is expressed upon the concurrent delivery of the inducer, although the expression is transient: the expression of cargo protein (TP) by pBAD lasted only 1 day after administration of L-arabinose into the peritoneal cavity (Appendix A). In this case, daily administration of L-arabinose would be required to prolong the expression. Another approach would be to use hypoxia-responsive promoters that are activated in tumor-colonizing bacteria [51]. Nevertheless, if the purpose of the controlled expression is to prevent toxic substances from harming healthy organs such as the liver and spleen, which are responsible for 60% and 30% of the immunological removal of bacteria from the circulation, respectively [52], such practice is no longer needed. Administration of *S.* Gallinarum constitutively expressing cytolysin A, a potent pore-forming hemolytic protein of *S. enterica* serovar Typhi [20], into tumor-bearing mice had no adverse effect on the animals. When an extension of anti-cancer cargo expression is needed, a multiple bacteria injection could be an option. In bacterial cancer therapy, the bacteria detected in the RES should be considered inert. In this study, we used the exponential phase promoter *rrnB* P1 to express TP in *S.* Gallinarum proliferating in tumor tissues exclusively, which conferred considerable antitumor effects without any systemic toxicity.

## 5. Conclusions

This study demonstrated that bacteria that reside in tumor tissues actively proliferate, whereas those in the RES die off after injection into tumor-bearing mice. A cytotoxic anticancer protein gene fused to a constitutive promoter was expressed only in the bacteria residing in the tumor tissue, resulting in tumor suppression.

## Figures and Tables

**Figure 1 cancers-15-01486-f001:**
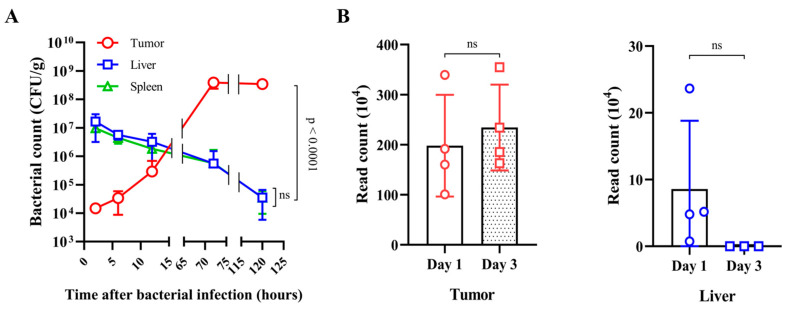
Fate of *E. coli* injected into tumor-bearing mice. BALB/c mice grafted with CT26 colon carcinoma cells received intravenous injection of *Escherichia coli* MG1655 (*n* = 4 per group, 1 × 10^8^ CFU/mouse) when tumor volumes reached 130–150 mm^3^. (**A**) At the indicated times after the injection, tumors, livers, and spleens were extracted to determine bacterial counts (CFU/gram). (**B**) RNA sequencing analysis was performed to determine the expression profile of the rrnB operon (number of reads) (NC_000913.3) in *E. coli* residing in tumors and livers at 1 and 3 days postinjection (dpi). (**C**) The rrnB P1 promoter sequence with three *Fis*-binding sites (I–III) in the upstream activation region. (**D**) *E. coli* MG1655 carrying prrnBP1-gfpOVA (EMP4002) was administered intravenously to CT26-grafted mice (*n* = 5/group, 1 × 10^8^ CFU/mouse). To monitor rrnB P1 promoter activities in vivo, the levels of *gfp* expression in tumors, livers, and spleens at each time point were determined by quantitative real-time PCR. The *E. coli* rpoB gene was used as a reference, and data were normalized by the 2^−ΔΔCt^ method. All data are expressed as the mean ± SD. *p*-values indicate differences between groups (unpaired Student’s *t*-tests; ns: not significant).

**Figure 2 cancers-15-01486-f002:**
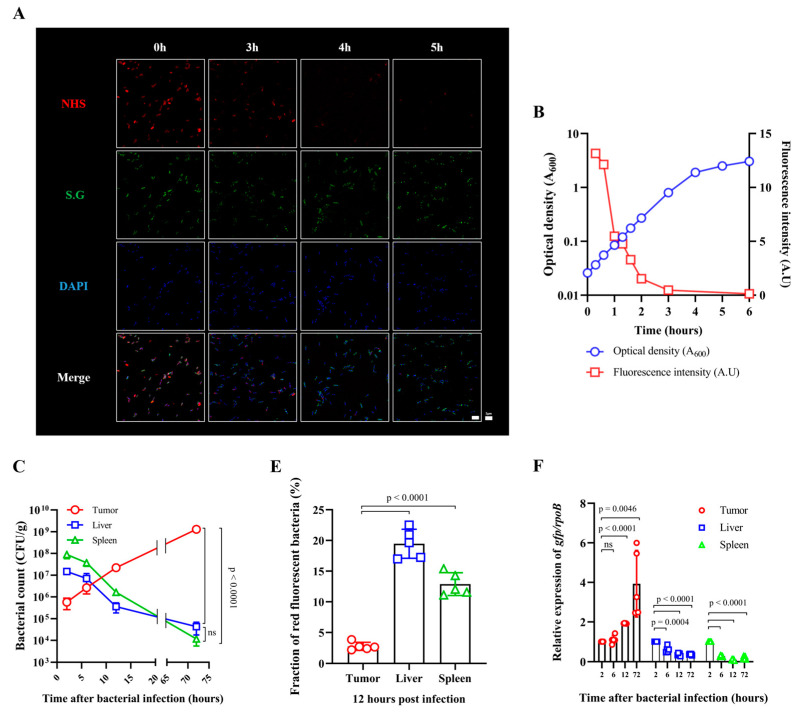
Fate of ΔppGpp *S.* Gallinarum injected into tumor-bearing mice. (**A**) Images of ΔppGpp *S.* Gallinarum cross-linked with a red fluorescent dye (Flamma^®^ Fluors 552 NHS ester) grown in LB medium for the indicated amounts of time. Bacteria stained with a fluorescent antibody are shown in green and DAPI-stained nuclei are shown in blue. Scale bar = 5 µm for 1000× magnification. (**B**) The red fluorescent bacteria were grown in LB with vigorous aeration. At the indicated time points, samples were collected to measure cell mass (A_600_) and red fluorescent signals using a fluorometer at an excitation wavelength of 550 nm and an emission wavelength of 610 nm. A.U. = red fluorescent signal/A_600_. (**C**) ΔppGpp *S.* Gallinarum was intravenously injected into CT26 tumor-bearing mice (*n* = 4/group, 1 × 10^8^ CFU/mouse) when tumor size reached 130–150 mm^3^. Bacterial loads from isolated tumors, livers, and spleens were determined at the indicated times after the injection (related to data shown in Appendix A). (**D**) The fluorescent bacteria were intravenously injected into mice grafted with CT26 cancer cells (*n* = 5, 1 × 10^8^ CFU/mouse). The Salmonella in tumors, spleens, and livers at the indicated time points were observed under a confocal microscope. Bacteria cross-linked with Flamma^®^ Fluors 552 NHS ester are shown in red, bacteria stained with a fluorescent antibody are shown in green, and DAPI-stained nuclei are shown in blue. Yellow arrows indicate merged red and green bacterial signals. Scale bar = 10 µm for 800× magnification. (**E**) The fractions of red fluorescent bacteria (indicated by arrows) among green bacteria (total number) in each organ and tumor tissue at 12 h postinjection are shown in a graph (fraction of red fluorescent bacteria = red/green × 100, %). Results are expressed as the mean ± SD from five sections, unpaired Student’s *t*-tests, *p* < 0.0001. (**F**) ΔppGpp *S.* Gallinarum carrying prrnBp1-gfpOVA (SMP4001) was administered to CT26-grafted mice (*n* = 5/group) via the tail vein, and gfpOVA expression from the rrnB P1 promoter in the bacteria in tumors, livers, and spleens at the indicated time points was determined by quantitative real-time PCR. The gfpOVA Ct values were calculated relative to *S.* Gallinarum rpoB Ct values in triplicate. All data are expressed as the mean ± SD. Significant differences are shown as *p* < 0.05 (unpaired Student’s *t*-tests; ns: not significant).

**Figure 3 cancers-15-01486-f003:**
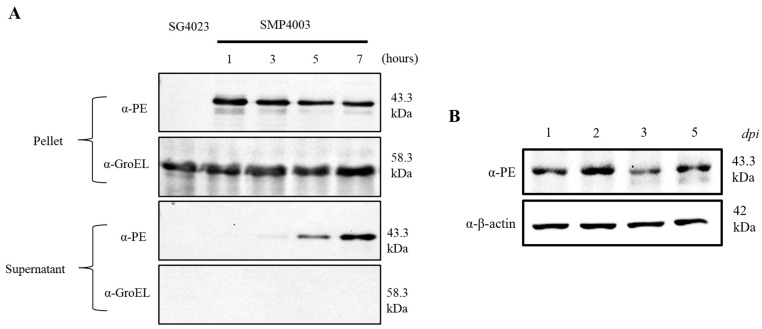
Expression and secretion of TGFα-PE38 (TP) from ΔppGpp *S.* Gallinarum. (**A**) Expression of TP in vitro. ΔppGpp *S.* Gallinarum carrying prrnBP1-psp-TP (SMP4003) were cultured in LB media, and samples were taken at the indicated time points. Bacterial samples were centrifuged to divide into supernatant and pellets. The pellets were sonicated. Aliquots containing 20 ng total protein were loaded onto 10% SDS-PAGE gels to detect TP (43.3 kDa) by western blotting using an antibody against *Pseudomonas* exotoxin A. The efficiency of bacterial lysis in the pellet was determined using GroEL (58.3 kDa) as a cytosolic protein control. The first lane contains SG4023 without the plasmid. Data represent the results of three independent replicates. Uncropped membranes are shown in panels A, B, and C in Appendix A. (**B**) Expression of TP in vivo. The above bacteria were injected into CT26 tumor-bearing mice via the tail vein (1 × 10^8^ CFU/mouse). On the indicated days, tumors were excised and homogenized, lysates were centrifuged and filtered, and supernatants were collected. The presence of TP was determined by western blotting. β-actin was used as the loading control (42 kDa). Representative data are the results of two independent replicates. Uncropped membranes are shown in panels D and E in Appendix A.

**Figure 4 cancers-15-01486-f004:**
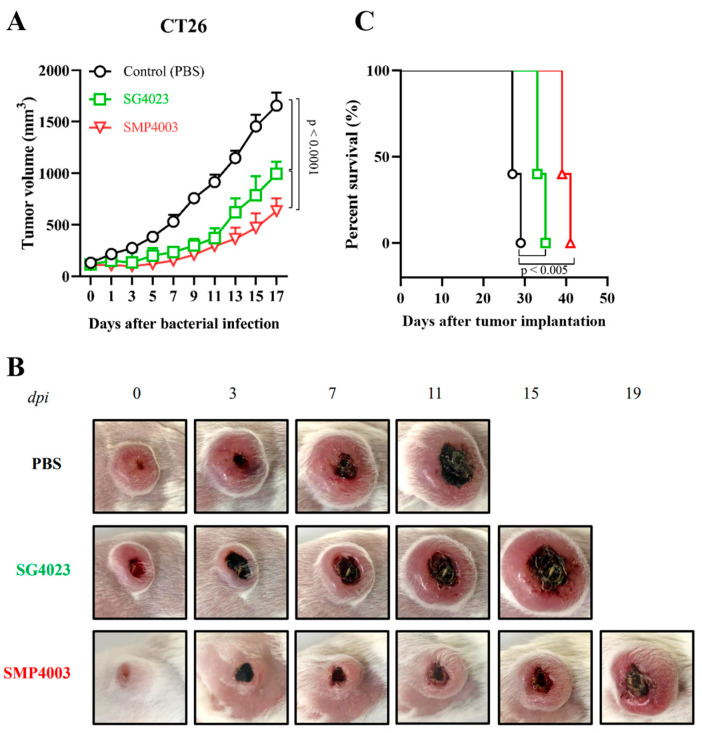
Antitumor effect of ΔppGpp *S.* Gallinarum carrying prrnBP1-psp-TP in BALB/c mice grafted with CT26 colon carcinoma and 4T1 murine breast cancer cells. BALB/c mice were subcutaneously implanted with 1 × 10^6^ CT26 colon carcinoma cells (*n* = 5 per group, (**A**–**C**)) or 1 × 10^6^ 4T1 murine breast carcinoma cells (*n* = 6 per group, (**D**–**F**)) on the high flank. Mice were treated with PBS (black line), ΔppGpp *S.* Gallinarum (SG4023, green line), or *S.* Gallinarum transformed with the therapeutic plasmid prrnBP1-psp-TP (SMP4003, red line) by intravenous injection at a dose of 1 × 10^8^ CFU/mouse when tumor volumes reached 90–120 mm^3^. Average tumor sizes in each group of mice grafted with CT26 (**A**) or 4T1 (**D**) were recorded every 2 days after treatment with the engineered bacteria until the volumes reached > 1500 mm^3^ (two-way ANOVA with Tukey’s multiple comparisons test, *p* < 0.0001). Representative images of CT26 carcinoma (**B**) or 4T1 carcinoma (**E**) in the above mice. The survival of CT26 tumor-bearing mice (**C**) or 4T1 tumor-bearing mice (**F**) treated as described above was determined using Kaplan–Meier curves (logrank Mantel–Cox test, *p* < 0.005 and *p* < 0.05 for (**C**,**F**), related to data shown in Appendix A.

**Table 1 cancers-15-01486-t001:** Bacterial strains and plasmids used in this study.

Strains	Description	References
*Escherichia coli*		
MG1655	Wild type (with defects in ilvG, rfb50, and rph-1)	ATCC [30,31]
EMP4002	MG1655, prrnBP1-gfpOVA, *Amp^r^*	This work
*Salmonella enterica* serovar Gallinarum		
SG4021	Wild-type isolate, clinical	
SG4023	SG4021, Δ*relA*, Δ*spoT*	[26]
SG4030	SG4023, Δ*relA*, Δ*spoT*, Δ*glmS::Kan^r^*	This work
SMP4001	SG4023, prrnBP1-gfpOVA, *Amp^r^*	This work
SMP4003	SG4030, prrnBP1-psp-TP, *glmS+, Amp^r^*	This work
SMP4004	SG4030, pSEC-TGFα-PE38, *glmS+, Amp^r^*	This work
Plasmids		
prrnBP1-gfpOVA	*gfpOVA* under control of P_rrnB P1_ in pBR322	This work
prrnBP1-psp-TP	psp-TP under control of P_rrnB P1_ in pBAD24	This work (Appendix A)
pSEC-TGFα-PE38	psp-TP under control of P_araBAD_ in pBAD24	[25]

**Table 2 cancers-15-01486-t002:** Specific primer sequences for engineered plasmids.

Construction	Name and Direction *	Sequence **
prrnBP1-gfpOVA	GFP-vector-FW	5′-CGGAATAACTCCCTATAATGCGCCACCACTTCTAGATTTAAGAAGGAGATATACATATGA-3′
	GFP-vector-RV	5′-AACGCTGTAAAACGGGCAATAATTGTTCAGCGCATGCACCATTCCTTGCGGCG-3′
	rrnB1-insert-FW	5′-CGCCGCAAGGAATGGTGCATGCGCTGAACAATTATTGCCCGTTTTACAGCGTT-3′
	rrnB1-insert-RV	5′-TCATATGTATATCTCCTTCTTAAATCTAGAAGTGGTGGCCATTATAGGGAGTTATTCCG-3′
	Seq-GFP-FW	5′-ATAAGTGCGGCGACGATAGTCAT -3′
prrnBP1-psp-TP	psp-vector-FW	5′AATAACTCCCTATAATGCGCCACCACTATGGGTTTGAAGATGAAGAAAAGATCAG-3′
	psp-vector-RV	5′-AACGCTGTAAAACGGGCAATAATTGTTCAGCCTCTGAATGGCGGGAGTATGAAAA-3′
	rrnB2-insert-FW	5′-CCATACTTTTCATACTCCCGCCATTCAGAGGCTGAACAATTATTG CCCGTTTTAC-3′
	rrnB2-insert-RV	5′-GCCTGATCTTTTCTTCATCTTCAAACCCATAGTGGTGGCGCATTATAGGG-3′
	Seq-psp-FW	5′-AAAATCGAGATAACCGTTGGCC-3′

* FW: forward primer; RV: reverse primer. ** The underlined nucleotides are the homologous sequences for the assembly.

## Data Availability

The experimental data are available on request by corresponding authors.

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
