# Peer review of "Constitutive Expression of a Cytotoxic Anticancer Protein in Tumor-Colonizing Bacteria"

_cancers, 2023, doi:10.3390/cancers15051486_

Round 1

Reviewer 1 Report

In their manuscript, Phuong-Thu Mai describe the construction of tumor-therapeutic bacteria and test their efficacy in vitro and in vivo. They first test a laboratory strain of E. coli for highly expressed genes in tumors and derive a promoter that controls the ribosome synthesis during exponential growth. They then attenuate S. Gallinarum by deleting the property to synthesize ppGpp and introduce an expression plasmid that encodes a fusion protein consisting of bacterial secretion motif as well as TGFa and an Exotoxin of P. aeruginosa. After characterization, they could show that this immunotoxin is active against two tumor cell lines in vitro and improves the therapeutic efficacy of S. Gallinarum against these two tumors in vivo. The authors have presented very interesting work and important data. The experiments are carefully carried out and well described. The manuscript deserves publication provided a few minor point are addressed.

1.      The authors describe a promoter that is constitutively expressed in tumor tissue for up to at least 5 days but only very transiently in spleen and liver after administration. They believe that this is due to the fact that the promoter which controls ribosome synthesis is active mainly in the log phase of bacterial proliferation which only takes place in the tumor tissue. However, they show that after 75 hours the bacteria in the tumor no longer increase in numbers and thus do not proliferate. How can they reconcile this findings.

2.      The plasmid pSEC-TGFa-PE28 is very essential to this work. A plasmid map should be provided in the supplement. The description of the elements is found in the results section except the SEC motif for which no information is provided in the text. This should be provided.

3.      MG1655 could be considered WT. Nervertheless, it bears mutations which might be important under these conditions. ilvG which is responsible for synthesis of branched AA; rfb50 which results in absence of O-antigen and might render the bacteria sensitive to complement and phagocytosis; rph-1 resulting in slight pyr auxotrophy. This should be added in Table 1.

4.      The authors carried out transcriptomic of tumor residing bacteria but mention only the rrnB unit. Why?

5.      Line 265: The authors mention that the rrnB activity in liver is due to remnants of the over night culture. I cannot understand. They would be in stationary phase. Thus little activity should occur. Despite their inertness in liver and spleen. Most bacteria at that time are still alive and might assemble ribosomes. Please clarify.

6.      Line 331: The dye is not “bleached” it is “diluted”. Please correct.

7.      The data of the fluorescence panels are very difficult to see. They should be improved. Especially very little qualified evidence is presented that the bacteria in liver are associated with macrophages. This also needs improvement.

8.      Lines 392-394. The correct control would be the supernatant from parental bacteria that do not carry the expression plasmid. This control was done but only shown in the figure mentioned later in the text. Please correct.

9.      In general, the indication (A, B….) on the various panels of figures are almost randomly placed. This is sometimes disturbing. Please place them systematically.

10.   In general, the experiments are apparently carried out only once although international standard is that results need at least reproduced once. However, in the present case two independent tumor systems were tested with similar results. Thus, it is acceptable.

11.   The authors mention inducible promotors used to activate gene expression in tumor residing bacteria but refer only to their own work. They should also cite Loessner et al. Cell. Microbiol. 2007 and Loessner et al. Microbes Infection 2009 since they used the mentioned compounds first.

Reviewer 2 Report

The paper by Mai P-T et al. have investigated the ability of tumor-colonizing bacteria to express a cytotoxic anticancer protein and exert anti-tumour growth effects in two cancer models CT26 and 4T1. This is an interesting and important study and the manuscript is well-written. However before it can be accepted for publication there are several points/amendments that need addressing:

1.In all the figures in the manuscript the labels (e.g. A, B, C) have been placed in random places next to the graph/image making it very hard to follow which figure is which. Please be consistent in your placing of the labels ie top left of the image/graph is normal. In figure 2 I was initially confused as there didn’t seem to be a figure 2D as this image did not follow Fig2C but comes after Fig 2F and is then not labelled with a D.

2.The authors use the data shown in Fig S2 to argue that the presence of macrophages associated with the bacteria in RES was responsible for the clearance of the bacteria in RES and not in tumor. Whilst the images do seem to support this they are very high magnification images showing only a small area and thus how representative are they of the total tissue area. Is it possible to quantify the total colocalization of F4/80 cells and bacteria across a larger tissue area in the tumour, liver and spleen. Also the authors have not considered the differences between tumour cells and the normal cellular response to sensing bacteria where the response of mutant tumor cells may be defective in triggering the activation of antimicrobial defences thus allowing bacterial proliferation.

3.In the in vivo experiments looking at the antitumor effect of the antitumor effect of ΔppGpp S. Gallinarum carrying prrnBP1-psp-TP in BALB/c mice 435 grafted with CT26 colon carcinoma and 4T1 murine breast cancer cells, it was notable that no complete tumour regressions were observed but rather just a retardation of tumour growth. What is the authors explanation for this? Did the authors check how long were the bacteria still present within the tumour? Also it was notable on the survival curves that the differences between the groups of mice (control vs SG4023 vs SMP4003) were not that great. I am also concerned by the stats on these survival graphs – the stats data for the  CT26 seems to show that even the control vs SG4023 had a significance of p<0.001 – this seems hard to believe based on the number of mice in each of the groups and the difference in actual survival of only a few days?? There is no mention in the Material and methods on what statistical test was used to determine the significance of the Kaplan -Meier curves. Normally a Log Rand (Mantel-Cox) analysis would be performed for this kind of data.

4. In  Figure 4 I think it is helpful to also show the average tumor size changes of CT26 and 4T1 alongside the individual group growth curves. These graphs are currently in Figure S4 A and C but would be better placed within Figure 4.

5.In the introduction the authors highlight that the initial response to the bacterial colonization of tumors is the secretion of the proinflammatory cytokine TNFα by innate immune cells, which causes a hemorrhage in the tumor and the formation of large necrotic regions. Do the authors  have any histological images of sections of the tumours from the different groups (PBS vs SG4023 vs SMP4003) showing the extent of necrotic areas? Did they also attempt to do any analysis other than F4/80 of immune infiltrate in these differently treated tumours ie CD8+ cytotoxic  T cells activated by the colonizing bacteria which as the authors stated themselves are the main type of cells that counteract tumor growth.
